# Individualized Triplet Chemotherapy Decision-Making in Metastatic Colorectal Cancer: A Machine-Learning-Driven Study [note 1]

**DOI:** 10.3390/cancers17223704

**Published:** 2025-11-19

**Authors:** Mehmet Kayaalp, Erman Akkuş, Beliz Bahar Karaoğlan, Güngör Utkan

**Affiliations:** 1Department of Medical Oncology, Faculty of Medicine, Ankara University, Ankara 06620, Türkiye; erman_akkus@yahoo.com (E.A.); bbaharulas@gmail.com (B.B.K.); 2Cancer Research Institute, Ankara University, Ankara 06620, Türkiye

**Keywords:** colorectal cancer, machine learning, FOLFIRINOX, triplet chemotherapy, individualized treatment

## Abstract

Managing metastatic colorectal cancer is challenging, and although intensive chemotherapy regimens such as FOLFOXIRI or FOLFIRINOX can provide substantial benefit, they are not appropriate for every patient. This study aimed to develop a tool that helps clinicians identify which individuals are most likely to benefit from these intensive treatments. Using data from 136 patients, the researchers built a model based on 10 key variables—including tumor sidedness and several routine blood biomarkers such as ferritin, CA19-9, and CRP—to predict which patients are more likely to experience longer periods without disease progression. This predictive approach may support more personalized treatment decisions by helping ensure that patients receive the most suitable therapy while avoiding unnecessary toxicity. Future prospective studies will be needed to validate the model’s clinical utility.

## 1. Introduction

This article is a revised and expanded version of a paper entitled “899eP—Individualised triplet chemotherapy decision-making in metastatic colorectal cancer: A machine learning–driven study,” which was presented at the ESMO Congress 2025, Berlin, Germany, 17–21 October 2025 [1].

First-line treatment of metastatic colorectal cancer primarily consists of 5-fluorouracil-based chemotherapy regimens in patients with non-resectable disease. Patients with potentially resectable or oligometastatic disease may instead undergo local treatment modalities, including radiotherapy, transarterial radioembolization (TARE), transarterial chemoembolization (TACE), or radiofrequency ablation (RFA), depending on disease extent and multidisciplinary evaluation. Oxaliplatin and irinotecan, when combined with fluorouracil, are used as doublet regimens in the first-line treatment of metastatic colorectal cancer, often in combination with anti-EGFR (epidermal growth factor receptor) or anti-VEGF agents. Although some studies have demonstrated a progression-free survival (PFS) and overall survival (OS) advantage of triplet therapy over doublet regimens, while others have found comparable outcomes, triplet therapy remains a more toxic option and should be used with caution, particularly in frail patient populations [2,3].

In pivotal clinical trials, triplet therapy has been shown to improve progression-free survival (PFS) from approximately 9 months to 12 months compared to irinotecan-based doublet regimens [4]. However, due to the absence of a clearly superior subgroup identified in post hoc analyses, current guidelines do not provide a definitive recommendation regarding the specific patient populations in whom triplet therapy should be initiated.

The ASCO guideline recommends triplet chemotherapy for patients in whom a rapid response is desired, while it does not prioritize this approach for those with poor performance status, significant comorbidities, advanced age, or MSI-H (microsatellite instability-high) tumors. Furthermore, the combination of triplet chemotherapy with anti-EGFR therapy is not recommended. However, these recommendations are based on evidence of low quality [5].

A recent pooled analysis of TRIBE, TRIBE2, CHARTA and STEAM showed that FOLFOXIRI plus bevacizumab offers superior PFS and higher response rates compared with doublets in selected high-risk groups—such as those with unresectable liver-only disease, right-sided tumors, or RAS/BRAFV600E mutations—independent of secondary resection [6]. Yet, the absence of validated biomarkers and inconsistent subgroup findings highlight the ongoing need for more precise, individualized approaches to guide triplet therapy selection.

Studies on circulating tumor DNA (ctDNA) have provided promising insights and potential clinical guidance. However, its integration into routine clinical practice remains limited due to unresolved concerns regarding cost-effectiveness [7].

Causal inference is the discipline of estimating the effects of interventions by using assumptions about the underlying data-generating process, since such causal questions cannot be answered from observational distributions alone [8]. In recent years, machine learning–based causal frameworks have increasingly been used to estimate heterogeneous and individual-level treatment effects from high-dimensional data. The Individual Treatment Effect (ITE) represents the difference between an individual’s potential outcome under treatment and the potential outcome for the same individual under control, quantifying how much benefit or harm a specific patient would experience from the intervention [9]. An illustrative example of this approach is a secondary analysis of the ASPREE randomized trial, which applied an effect score model to distinguish subgroups of older adults who were more likely to benefit from, or be harmed by, low-dose aspirin in terms of cancer risk; a finding that highlights how causal machine learning methods can uncover treatment-responsive subpopulations that conventional average-effect analyses fail to identify [10]. Meta-learners address the fundamental missing-counterfactual problem by first estimating the conditional outcome functions μ_0_(x) and μ_1_(x), and then imputing the individual treatment effects that cannot be directly observed [11].

Current literature shows that oncologic treatment decisions are predominantly guided by average treatment effects derived from randomized clinical trials. In the personalization of cancer therapy, appropriately controlling for confounders, understanding biological mediator mechanisms that shape treatment response, and minimizing common sources of selection bias are essential for improving the reliability of causal inference–based models. Patients encountered in real-world clinical practice often differ substantially from RCT populations, and treatment responses are inherently heterogeneous. Consequently, approaches capable of estimating the Individual Treatment Effect (ITE) are increasingly needed [12]. Methods developed using randomized trial data have shown that meta-learner and causal forest frameworks can successfully quantify such heterogeneity, enabling more individualized treatment decision-making [13].

Machine learning methods have been successfully used to support treatment selection and predict survival in various solid tumors, including hepatocellular carcinoma [14]. An XGBoost-based model that predicted 6-month mortality before treatment initiation in advanced cancer patients demonstrated high accuracy using a limited set of clinical parameters, highlighting the potential of such approaches to inform hospice-related decision-making [15]. Correlation-based models, however, fail to accurately capture true treatment effects in the presence of confounding [16]. The lack of reliable individualized effect estimates is particularly critical in treatment settings where toxicity is substantial and optimal patient selection remains unclear, such as triplet chemotherapy in metastatic colorectal cancer.

Therefore, this retrospective study, based on the Ankara University Colorectal Cancer Database, was designed to develop a machine learning–based model aimed at predicting which patients are likely to experience a progression-free survival (PFS) benefit from triplet chemotherapy. By identifying individuals who are most likely to benefit from this more intensive but potentially more toxic treatment approach, the study aims to support personalized first-line treatment decision-making in patients with metastatic colorectal cancer.

## 2. Materials and Methods

A total of 136 patients were included in this study, all registered in the Ankara University Faculty of Medicine Synchronous Metastatic CRC Cohort (AUTF-DMKRK) with a diagnosis of de novo metastatic colorectal cancer and complete treatment information available.

All data were extracted from the Avicenna Hospital Data Management System and entered into the institutional colorectal cancer registry. Patient records are routinely updated by attending physicians at each visit. The data cut-off date for this study was February 2024.

For each patient, 66 features were collected, including:Demographic data (gender, age at diagnosis),Comorbidities (diabetes, hypertension, coronary artery disease, hypothyroidism),Lifestyle information (smoking status),Family history (presence of cancer in first-degree relatives),Tumor characteristics (primary localization, histological subtype—mucinous; molecular markers—MSI, RAS, RAF),Sites of metastatic involvement (liver, peritoneum, lung, bone),Need for emergency surgery, primary tumor resection status,Comprehensive laboratory parameters (CEA, CA19-9, LDH, thyroid function tests, lipid panel, complete blood count, liver and kidney function tests, vitamin D, B12, folate, ferritin, zinc),Performance status (ECOG),Anthropometric measurements (height, weight),Local treatment information (surgery or radiofrequency ablation).

Analyses were conducted using Python v3.12 in combination with open-source libraries, including scikit-learn and SHAP.

All variables were converted into numerical form, and missing values were handled using the “coerce” method. The target variable was progression-free survival (PFS) measured in days. Treatment status was represented by a binary variable named “triplet,” where a value of 1 indicated patients who received triplet chemotherapy (FOLFOXIRI, FOLFIRINOX, or mFOLFIRINOX), and 0 indicated those who did not.

The individual treatment effect was estimated using the T-Learner method, in which separate regression models were trained for patients who received triplet chemotherapy and for those who did not. Both models were implemented using the HistGradientBoostingRegressor algorithm. For each test patient, predictions were obtained from both models (μ_1_: prediction for the triplet-treated scenario, μ_0_: prediction for the untreated scenario), and the individual treatment effect was calculated asITE = μ_1_ − μ_0_.

Due to the limited number of patients, the data were not split into separate train and test sets. Therefore, the Leave-One-Out Cross-Validation (LOOCV) method was employed while measuring the performance. In each iteration, one patient was designated as the test sample, and the treatment arm models were retrained using the remaining data. This process was repeated for all patients, and ITE predictions were calculated accordingly. To perform SHAP analysis, a meta-model over the entire dataset was trained. Based on the mean absolute SHAP values, the top 5, 7, and 10 most impactful features were identified.

The T-Learner approach was subsequently retrained using only the most significant features identified by SHAP, and ITE predictions were recalculated for each patient using LOOCV. The performance was evaluated by comparing the predicted treatment effects to actual treatment assignments, limited to patients with progression-free survival (PFS) greater than 270 days. Discriminative ability was assessed using the receiver operating characteristic (ROC) curve and the area under the curve (AUC) metric.

For model implementation, the T-Learner consisted of two scikit-learn HistGradientBoostingRegressor models trained separately in the triplet and control arms. We prioritized stability and interpretability and therefore did not perform an exhaustive hyperparameter search; primary analyses used the library defaults (learning rate = 0.1; squared-error loss; 100 boosting iterations; default tree/leaf settings and early-stopping behavior). Sensitivity analyses with constrained settings (e.g., max_depth = 3–4, max_iter = 50–100) yielded consistent results. Features were coerced to numeric; missing values were handled as in the code (mean imputation where applied).

## 3. Results

The demographic characteristics of the patients and the treatments they received are presented in Table 1.

A total of 136 patients diagnosed with de novo metastatic colorectal cancer were included in the study. Using the T-Learner approach, the individual treatment effect (ITE) of triplet therapy on progression-free survival (PFS) was estimated for each patient. Model generalizability was evaluated using Leave-One-Out Cross-Validation (LOOCV).

Model performance was evaluated in a subset of patients with PFS longer than 270 days. ROC analysis demonstrated that the model trained with all features had the highest discriminative performance (AUC = 0.919), followed by models using the top 10 features (AUC = 0.869), top 7 features (AUC = 0.864), and top 5 features (AUC = 0.780) (Figure 1).

A meta model over the entire dataset was trained which was followed by SHAP analysis. According to the SHAP summary plot, the top 10 features with the highest impact on model output were identified as follows: primary tumor localization, ferritin, CA19-9, CRP, uric acid, TSH, triglyceride, total protein, LDL, and platelet (plt) (Figure 2).

These findings suggest that SHAP-derived feature selection preserves a substantial portion of predictive power and that reasonable estimation of individual treatment effect can be achieved using a limited number of key variables.

## 4. Discussion

This study represents, to best our knowledge, the first attempt to develop a machine learning model capable of predicting which patients with colorectal cancer are likely to benefit from triplet chemotherapy.

Triplet therapy is known to offer potential benefits in terms of progression-free survival (PFS) and overall survival (OS) in the first-line treatment of metastatic colorectal cancer [2]. However, current clinical guidelines do not provide clear recommendations regarding which patient subgroups should receive this regimen. Additionally, due to its high toxicity, clinicians do not consistently favor triplet over doublet chemotherapy.

Moreover, it is crucial to recognize that triplet chemotherapy does not consistently yield superior outcomes across all patient groups [3]. For example, the HORG trial found no statistically significant improvement in overall survival with triplet therapy compared to standard doublet regimens [4].

The advantage of the T-Learner model lies in its ability to estimate individualized treatment effects (ITE), as well as allowing to identify specific patient characteristics predictive of greater benefit from intensive treatment. These characteristics could serve as practical biomarkers or clinical decision-making tools. While some patients clearly benefit from triplet therapy, others—particularly those with impaired performance status or comorbidities—may experience disproportionate toxicity without a meaningful survival advantage. This underscores the need for careful clinical judgment when selecting the intensity of treatment.

A meta-analysis including five clinical trials comparing doublet and triplet chemotherapy found no significant differences among subgroups that could predict the efficacy of triplet regimens. The only exception was observed in patients with BRAF-mutant tumors, where the efficacy of the FOLFOXIRI plus bevacizumab regimen was found to be different from that of doublet regimens [5]. Consistent with these findings, a pooled analysis of the TRIBE, TRIBE2, CHARTA, and STEAM trials showed that triplet therapy provides a progression-free survival and response rate advantage in selected high-risk groups, although no validated predictive subgroup has been definitively established [6].

Our study aimed to develop a machine learning–based model to identify subgroups of patients with metastatic disease who may benefit from triplet chemotherapy, using the median progression-free survival of 9 months achieved with doublet regimens as a reference point.

The primary tumor location emerged as the most influential parameter in the established paradigm, suggesting that triplet chemotherapy may be more effective in left-sided colon cancers. Triplet mFOLFIRINOX is used as an induction regimen in patients with locally advanced and metastatic rectal cancer [7]. However, meta-analyses comparing doublet and triplet regimens have not demonstrated a significant difference between right- and left-sided colon cancers. Although the greater efficacy of triplet regimens in left-sided colon tumors may be a statistical coincidence, it was identified as the most important parameter in our model and should be further investigated in clinical studies. It is particularly noteworthy that, in our study, the triplet treatment arm—although not combined with anti-EGFR agents, which are not routinely recommended or used—demonstrated greater efficacy in the subgroup of patients with left-sided tumors.

Given that elevated CA19-9 is a marker of poor prognosis [17] in metastatic colorectal cancer, our results imply that the use of triplet chemotherapy in this high-risk subgroup may offer a greater therapeutic benefit compared to standard regimens. In our cohort, triplet chemotherapy was consistently administered in combination with bevacizumab. However, the observed association between elevated CA19-9 and predicted benefit from triplet therapy should be interpreted with caution. As bevacizumab was uniformly administered in the triplet arm, it is possible that the model captured an indirect effect related to anti-VEGF (vascular endothelial growth factor) activity [18] rather than a direct mechanistic relationship between CA19-9 and triplet chemotherapy. Alternatively, this finding may reflect sample-specific variation or a statistical coincidence, and thus requires validation in larger, prospective cohorts.

Uric acid, one of the top five parameters with the highest SHAP values in our model, has been reported in the literature to be associated with colorectal cancer development, BRAF mutation, and shorter progression-free survival (PFS) in non-metastatic colorectal cancer patients receiving doublet chemotherapy [19,20,21,22].

Triglyceride levels, identified by our model as having high SHAP value, have been reported to increase in FOLFOXIRI-resistant colorectal cancer cells, suggesting a potential link with reduced chemotherapy response [23].

Although the relationship between thyroid function and colorectal cancer or treatment response remains unclear, previous studies have identified a negative correlation between hypothyroidism and the risk of rectal cancer, whereas hyperthyroidism has been associated with an increased risk of colon cancer [24]. The observed association between low TSH levels and increased predicted benefit from triplet therapy may be attributed to the predominance of non-rectal tumors in our cohort.

Among the variables evaluated by our model, platelet count emerged as one of the most informative predictors of progression-free survival. While platelet levels are not commonly integrated into standard treatment decision algorithms, this finding is supported by growing evidence that links elevated platelet counts to tumor-associated inflammation and a more aggressive biological phenotype in colorectal cancer [25,26]. In the context of triplet chemotherapy regimens such as FOLFOXIRI, this inflammatory and pro-thrombotic milieu may attenuate the cytotoxic impact of treatment, contributing to suboptimal responses despite intensified therapy [27]. Although ferritin is a key marker of iron deficiency anemia, it also functions as an acute-phase reactant and may be elevated in cancer patients. Previous studies have demonstrated that elevated ferritin levels are associated with relapse in metastatic colorectal cancer. In our model, lower ferritin levels were found to be associated with a better response to triplet chemotherapy.

Many machine learning–based studies in the current literature attempt to guide clinical decision-making using limited data sources and correlation-driven approaches [28]; however, our work distinguishes itself by integrating routine clinical and laboratory variables within a causal machine learning framework aimed at estimating individualized treatment effects [29]. Causal inference methods and meta-learning models such as T-Learner have been used in both observational data and randomized controlled trials to estimate individualized treatment effects (ITEs); these approaches enable personalized and data-driven clinical decisions by comparing expected outcomes under different treatment scenarios for each patient [30].

T-Learner-based deep learning models have been utilized to support clinical decision-making by identifying optimal treatment strategies—specifically, to guide decisions between surgery and active surveillance in prostate cancer, to assess the impact of radiotherapy versus chemoradiotherapy on survival in glioblastoma patients, and to determine which patients with low-grade glioma are more likely to benefit from a given treatment option [31,32,33].

This study represents an initial step toward developing an interpretable machine learning framework to predict individualized benefit from triplet chemotherapy in metastatic colorectal cancer. While the model demonstrated promising discriminative performance using routinely available clinical and laboratory variables, its findings should be interpreted as hypothesis-generating rather than definitive. Given the heterogeneity of real-world treatment responses and the toxicity burden associated with intensive regimens, identifying patients most likely to benefit from triplet therapy remains a critical clinical need. Our results suggest that integrating data-driven individualized predictions with established clinical assessments may support more balanced treatment decisions. However, external and prospective validation is essential to confirm the model’s generalizability and real-world utility. Accordingly, future work will focus on designing a prospective, multicenter study to evaluate the framework’s performance and its potential role in guiding treatment selection in routine oncology practice.

## 5. Limitations

This study has several limitations. The individual predictive performance of the T-Learner’s base regressors (μ_0_ and μ_1_)—such as MSE, MAE, or R^2^—was not separately evaluated, as they were trained solely on routinely collected clinical variables. Consequently, the accuracy of these underlying regression models cannot be fully characterized beyond the overall ROC/AUC metrics.

The overall number of patients was only partially sufficient, which may influence the strength of the conclusions despite the use of LOOCV. The imbalance between patients receiving triplet versus non-triplet chemotherapy and the unequal distribution of anti-EGFR and anti-VEGF agents reflect real-world treatment patterns but remain inherent limitations of the retrospective design. In addition, the absence of immunotherapy use in patients with MSI-H tumors—due to reimbursement restrictions during the study period—represents another limitation. These factors should be taken into account when interpreting the findings.

## 6. Conclusions

Machine learning models are expected to play a significant role in guiding personalized treatment selection in oncology in general, and in colorectal cancer in particular. These models have the potential to improve personalized treatment strategies by integrating easily accessible, routinely collected categorical, clinical, and laboratory data available for most patients, thereby facilitating data-driven decision-making in real-world clinical settings. The model developed in this study may help address this unmet need, provided that its performance is validated in prospective clinical studies.

## Figures and Tables

**Figure 1 cancers-17-03704-f001:**
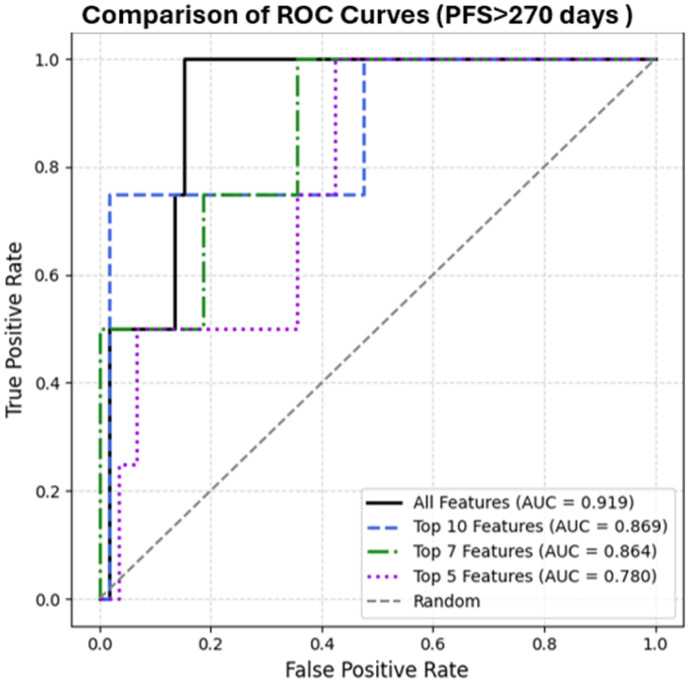
Comparison of ROC Curves Between Models.

**Figure 2 cancers-17-03704-f002:**
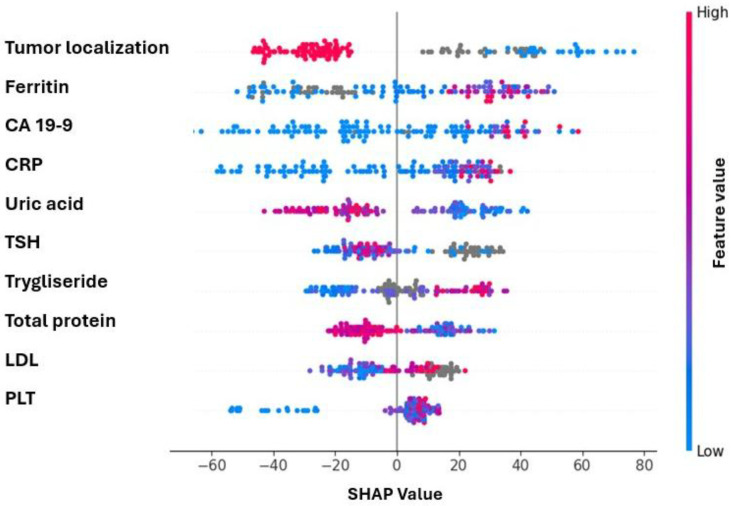
Comparison of ROC curves between the full model and SHAP-selected top 10 feature model using LOOCV.

**Table 1 cancers-17-03704-t001:** Patient demographics and clinical features.

Gender *n* (%)	Male 84 (61.8) Female 52 (38.2)
**meanAge(d)**	59.27 (10.71)
**Comorbidity n (%)**	Overall 84 (61.8) Diabetes 25 (18.4) Hypertension 42 (30.9) Hypothyroidism 9 (6.6) Coronary artery disease 13 (9.6)
**Smoking Status n (%)**	Never 70 (51.5) Ex-smoker 39 (28.6) Current 27 (19.8)
**Primary Site n (%)**	Right 27 (19.9) Left 78 (57.4) Not determined 31 (22.8)
**Metastatic** **Site n (%)**	Liver 120 (82) Peritoneal 18 (13.2) Lung 35 (25.7) Bone 6 (4.4)
**Mutation**	RAS Mutant 55 (40.4) RAF Mutant 3 (2.2)
**Emergent Surgery n (%)**	Performed 21 (15.4)
**Chemotherapy n (%)**	Single 2 (1.5) Doublet 128 (94.1) Triplet 6 (4.4)

## Data Availability

The data presented in this study are available on reasonable request from the corresponding author. The data are not publicly available due to privacy and ethical restrictions involving patient information.

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
