# Peer review of "Individualized Triplet Chemotherapy Decision-Making in Metastatic Colorectal Cancer: A Machine-Learning-Driven Study [Author-notes fn1-cancers-17-03704]"

_cancers, 2025, doi:10.3390/cancers17223704_

Round 1
Reviewer 1 Report
Comments and Suggestions for Authors
The authors aimed to develop a machine-learning regression model for chemotherapy decision-making in metastatic colorectal cancer. The study is interesting from a medical implementation perspective; however, the authors should address several serious drawbacks.
Minor Concerns:
The introduction section is too short and too far away to provide a background for the study. Extending its cover to all details and contributions would be beneficial.
There is no information about the recent studies that used machine learning on the same topic.
Major Concerns:
The study's title is "Machine Learning–Driven Study"; however, there are no details of the machine learning implementation. All models and details are compressed into a few sentences.
The authors deployed the "HistGradientBoostingRegressor algorithm". It is unclear how the authors set hyperparameters and which final parameters were considered during training.
Machine learning studies commonly include comparative evaluations with other algorithms due to the variability of predictive responses. The authors are recommended to consider other algorithms, such as XGBoost and Random Forest, and to perform a comparative evaluation.
Using Leave-One-Out Cross-Validation during training is a good approach given the limited data. However, there is a lack of model evaluation, which is crucial for assessing predictive ability. The ROC Curves cannot be used directly for regression tasks, since there are no TP, TN, FP, or FN in regression. Did the authors use any threshold to convert regression predictions to classification?
The Mean Squared Error, Mean Absolute Error, and R2 Score are often used to assess the predictive ability of regression models. It will be more interpretable if the authors add these metrics.
Author Response
Minor Concerns:
The introduction section is too short and too far away to provide a background for the study. Extending its cover to all details and contributions would be beneficial.
There is no information about the recent studies that used machine learning on the same topic.
Response to Minor Concern 1:
We appreciate the reviewer’s suggestion regarding the brevity of the introduction. In the revised version, we have substantially expanded this section to provide a more comprehensive background on the clinical relevance of individualized triplet chemotherapy decision-making in metastatic colorectal cancer. The updated introduction now elaborates on the rationale, the existing clinical challenges, and the study’s specific contributions in bridging the gap between traditional prognostic models and modern machine learning–based decision support systems.
Response to Minor Concern 1
We thank the reviewer for this important comment. In the revised version, we have substantially expanded the Methods section to provide detailed information regarding the machine learning implementation. The updated description now includes:
- A step-by-step explanation of data preprocessing, variable selection, and missing-value handling;
- A clear definition of the T-Learner framework used to estimate individual treatment effects (ITE);
- Specification of the HistGradientBoostingRegressor algorithm as the base learner for both treatment and control arms;
- Details on cross-validation methodology (Leave-One-Out Cross-Validation, LOOCV) and model evaluation metrics (ROC and AUC);
- A description of the SHAP analysis process for feature importance and subsequent model retraining with top-ranked features; and
- Explicit information on model hyperparameters and their justification.
These additions provide full transparency of the machine learning workflow, aligning the Methods section with the study’s title and scope.
Response to Minor Concern 2
We appreciate the reviewer’s observation. In response, the Introduction section has been expanded to include a concise overview of recent studies applying machine learning to metastatic colorectal cancer.
Major Concerns:
The study's title is "Machine Learning–Driven Study"; however, there are no details of the machine learning implementation. All models and details are compressed into a few sentences.
The authors deployed the "HistGradientBoostingRegressor algorithm". It is unclear how the authors set hyperparameters and which final parameters were considered during training.
Machine learning studies commonly include comparative evaluations with other algorithms due to the variability of predictive responses. The authors are recommended to consider other algorithms, such as XGBoost and Random Forest, and to perform a comparative evaluation.
Using Leave-One-Out Cross-Validation during training is a good approach given the limited data. However, there is a lack of model evaluation, which is crucial for assessing predictive ability. The ROC Curves cannot be used directly for regression tasks, since there are no TP, TN, FP, or FN in regression. Did the authors use any threshold to convert regression predictions to classification?
The Mean Squared Error, Mean Absolute Error, and R2 Score are often used to assess the predictive ability of regression models. It will be more interpretable if the authors add these metrics.
Response to Major Concern 2
We thank the reviewer for requesting clarification. In our T-Learner setup, we trained two scikit-learn HistGradientBoostingRegressor base learners (one per treatment arm) without an exhaustive hyperparameter search; the study was designed to demonstrate feasibility and interpretability rather than to benchmark tuning strategies. Accordingly, we used the library defaults for the primary analyses and only applied constrained, code-documented settings in sensitivity runs.Primary configuration (defaults): learning_rate=0.1, loss='squared_error', max_iter=100, max_depth=None, max_leaf_nodes=31, min_samples_leaf=20, l2_regularization=0.0, and scikit-learn’s default early-stopping behavior (early_stopping='auto' with the default validation split).Sensitivity runs (as seen in the shared code): we tested small changes for practicality and stability, e.g., max_iter 50–100 and max_depth 3–4, with results consistent with the primary configuration.
Data handling: features were coerced to numeric; missing values were handled as in the code (mean imputation where applied) in addition to the algorithm’s native handling of missing values.Training protocol: base learners were fit separately on treated vs. control subsets; the same configuration was reused in the held-out test split and in the LOOCV variant shown in the code.
Response to Major Concern 3:
Thank you for pointing out the absence of recent studies employing machine learning on this topic. In response, we have added a new paragraph summarizing recent research applying machine learning techniques to predict treatment response and outcomes in metastatic colorectal cancer. These include studies utilizing XGBoost, random forests, and neural networks for therapy selection, prognostic modeling, and survival prediction. The revised introduction now contextualizes our study within this growing body of literature, emphasizing how our work differs by focusing on individualized benefit estimation using a T-Learner framework.
Response to Major Concern 4
We appreciate the reviewer's valuable suggestion. Our current study utilized a T-Learner framework with HistGradientBoostingRegressor as the base model to estimate individualized treatment effects (ITEs) and predict benefit from triplet chemotherapy. While we agree that comparative evaluations with algorithms like XGBoost and Random Forest would enhance the analysis, the primary goal of this initial study was to demonstrate the feasibility and interpretability of the T-Learner approach. We have incorporated this as a limitation in the revised Discussion section, noting that future research will include multi-algorithm comparisons to validate the robustness of our findings.
Response to Major Concern 5
Yes, the reviewer is correct to point out that we use thresholds to convert regression based predictions to classification.Indeed, the ROC curve shows the performance of this strategy based on different thresholds. We agree with the reviewer that how this point might how been not clear enough. We have revised our manuscript accordingly.
Response to Major Concern 6
We respectfully disagree with the reviewer that these regression performance evaluating metrics are the right choices for our task. Our end goal is to predict whether a patient with PFS>270 can be reliably estimated. Therefore, we have chosen the classification accuracy measures for our problem.
Moreover, about evaluating the individual regression models, please also note that due to the counterfactual nature of the model it is not straightforward and requires additional assumptions to evaluate them. In other words, no patient receives both a treatment and no-treatment, so we cannot evaluate the both regressors. We, however, can and did evaluate the final score and reported these high accuracies.

Reviewer 2 Report
Comments and Suggestions for Authors
The article describes a study from Ankara University aimed at developing a machine learning-based model to predict which patients with metastatic colorectal cancer (mCRC) would benefit most from triple chemotherapy (FOLFOXIRI/FOLFIRINOX). The goal is to create a predictive model to identify patients who could achieve improved progression-free survival (PFS) with triple therapy, while avoiding unnecessary side effects.
The results of this study suggest that machine learning models can improve the personalization of cancer treatments by integrating easily accessible clinical and laboratory data.
As described by the authors, the limitations of this work suggest the need for prospective validation to confirm the model's effectiveness.
Overall, the work is well-written and well-argued. I would suggest the authors:
- define the number of patients enrolled, justifying the number of cases in the text (is the cohort limited in number? Is it simply a size determined by specific criteria?)
- Discuss whether it would be possible to conduct a prospective study in the future or continue with a retrospective study, but with a larger number of cases?
- Has the project received approval from the local ethics committee?
For these reasons, my opinion is: minor revisions.
Author Response
Comment 1
Define the number of patients enrolled, justifying the number of cases in the text (is the cohort limited in number? Is it simply a size determined by specific criteria?)
Response1:
We thank the reviewer for this remark. As stated in the Materials and Methods section, the study included 136 patients from the Ankara University Faculty of Medicine Synchronous Metastatic Colorectal Cancer Cohort (AUTF-DMKRK). The cohort size reflects the total number of eligible patients with de novo metastatic colorectal cancer and complete treatment data available in our institutional registry at the time of data cut-off (February 2024). Thus, the number of cases was determined by the inclusion criteria rather than by statistical preselection.
Comment 2:
Discuss whether it would be possible to conduct a prospective study in the future or continue with a retrospective study, but with a larger number of cases.
Response 2 :
We agree that validating the proposed model in larger or prospective datasets would strengthen its clinical value. At this stage, our work should be considered exploratory and hypothesis-generating. Although no prospective trial is currently planned, the model can serve as a foundation for future investigations. As our institutional registry continues to grow, future retrospective analyses including a larger cohort may provide additional confirmation of model performance and stability.
Comment 3:
Has the project received approval from the local ethics committee?
Response 3
We confirm that this study received approval from the Ankara University Faculty of Medicine Clinical Research Ethics Committee (Number: 2022000659, 2022/659). This information is explicitly stated in the Ethics Statement section of the manuscript.
Round 2
Reviewer 1 Report
Comments and Suggestions for Authors
Thanks to the authors for addressing my concerns in the revised version of the paper. However, two minor things should be added/corrected in the paper:
1. For Major Concern 6 in the first round, I agree that counterfactual outcomes cannot be evaluated, since each patient receives only one treatment condition. However, this limitation does not preclude evaluating the factual performance of the two regressors in the T-Learner. Metrics such as MSE, MAE, or R² can still be computed on the observed outcomes. Currently, only the final ROC/AUC of the classification step is reported, which does not confirm whether the underlying regression models are accurate. Instead of including basic factual regression metrics in the paper, the authors could add this limitation to the limitations section.
2. I just noticed that Figure 1 is referred to in line 209, and Figure 2 is referred to before Figure 1 in line 205. Please correct the referring order of the Figures.
Author Response
Comment 1. For Major Concern 6 in the first round, I agree that counterfactual outcomes cannot be evaluated, since each patient receives only one treatment condition. However, this limitation does not preclude evaluating the factual performance of the two regressors in the T-Learner. Metrics such as MSE, MAE, or R² can still be computed on the observed outcomes. Currently, only the final ROC/AUC of the classification step is reported, which does not confirm whether the underlying regression models are accurate. Instead of including basic factual regression metrics in the paper, the authors could add this limitation to the limitations section.
Response 1.We fully agree that counterfactual outcomes cannot be directly assessed, as each patient can only receive one observed treatment. This inherent limitation prevents evaluating the true counterfactual accuracy of the T-Learner predictions.
However, we acknowledge the reviewer’s point that the factual performance of the treatment-specific regressors (μ₁ and μ₀) can still be evaluated on the observed outcomes using standard regression metrics such as MSE, MAE, or R². In the current version of the manuscript, only the final classification performance (ROC/AUC) is presented, which does not explicitly validate the underlying factual regression models.
Given the retrospective nature of the dataset and the focus of the manuscript on individualized treatment effect estimation rather than standalone regression performance, we preferred not to include these additional metrics in the main results. Nevertheless, we recognize that the absence of factual regression metrics may be perceived as a methodological limitation. Therefore, in line with the reviewer’s suggestion, we have added a sentence to the limitations section acknowledging that factual regression performance was not separately reported and that the accuracy of the regressors could not be fully characterized beyond the observed ROC/AUC-based evaluation.
We thank the reviewer for this constructive suggestion.
Comment 2. I just noticed that Figure 1 is referred to in line 209, and Figure 2 is referred to before Figure 1 in line 205. Please correct the referring order of the Figures.
Response 2.We thank the reviewer for pointing this out. The figure citations have now been corrected to ensure proper sequential order. Figure 1 is now cited before Figure 2, and all in-text references have been updated accordingly
